# Novel Model to Predict HCC Recurrence after Liver Transplantation Obtained Using Deep Learning: A Multicenter Study

**DOI:** 10.3390/cancers12102791

**Published:** 2020-09-29

**Authors:** Joon Yeul Nam, Jeong-Hoon Lee, Junho Bae, Young Chang, Yuri Cho, Dong Hyun Sinn, Bo Hyun Kim, Seoung Hoon Kim, Nam-Joon Yi, Kwang-Woong Lee, Jong Man Kim, Joong-Won Park, Yoon Jun Kim, Jung-Hwan Yoon, Jae-Won Joh, Kyung-Suk Suh

**Affiliations:** 1Department of Internal Medicine and Liver Research Institute, Seoul National University College of Medicine, Seoul 03080, Korea; 83187@snuh.org (J.Y.N.); chyoung@schmc.ac.kr (Y.C.); presh_yuri@hanmail.net (Y.C.); yoonjun@snu.ac.kr (Y.J.K.); yoonjh@snu.ac.kr (J.-H.Y.); 2DEEPNOID Inc., Seoul 08376, Korea; ghost9023@deepnoid.com; 3Department of Internal Medicine, CHA Gangnam Medical Center, CHA University School of Medicine, Seoul 06135, Korea; 4Department of Internal Medicine, Samsung Medical Center, Sungkyunkwan University School of Medicine, Seoul 06351, Korea; sinndhn@hanmail.net; 5Center for Liver Cancer, National Cancer Center, Goyang-Si 10408, Gyeonggi-Do, Korea; bohkim@ncc.re.kr (B.H.K.); kshlj@hanmail.net (S.H.K.); jwpark@ncc.re.kr (J.-W.P.); 6Department of Surgery, Seoul National University College of Medicine, Seoul 03080, Korea; gsleenj@hanmail.net (N.-J.Y.); kwleegs@snuh.org (K.-W.L.); kssuh@snuh.org (K.-S.S.); 7Department of Surgery, Samsung Medical Center, Sungkyunkwan University School of Medicine, Seoul 06351, Korea; jongman94@hanmail.net (J.M.K.); jw.joh@samsung.com (J.-W.J.)

**Keywords:** deep learning, liver transplantation, hepatocellular carcinoma, Milan criteria

## Abstract

**Simple Summary:**

Although several models have been developed to extend the criteria for liver transplantation in hepatocellular carcinoma beyond the Milan criteria, there are still no standard criteria. This study aimed to develop and validate a novel model to predict hepatocellular carcinoma recurrence after liver transplantation by adopting artificial intelligence (MoRAL-AI). The MoRAL-AI showed significantly better discrimination (c-index = 0.75) than previous models in the independent validation cohort: the Milan (c-index = 0.64), MoRAL (c-index = 0.69), UCSF (c-index = 0.62), up-to-seven (c-index = 0.50), and Kyoto (c-index = 0.50) criteria (all *p <* 0.001). We assessed the weighted parameters for tumor recurrence in the MoRAL-AI with the deep learning method: tumor diameter, followed by alpha-fetoprotein, age, and PIVKA-II.

**Abstract:**

Several models have been developed using conventional regression approaches to extend the criteria for liver transplantation (LT) in hepatocellular carcinoma (HCC) beyond the Milan criteria. We aimed to develop a novel model to predict tumor recurrence after LT by adopting artificial intelligence (MoRAL-AI). This study included 563 patients who underwent LT for HCC at three large LT centers in Korea. Derivation (*n* = 349) and validation (*n* = 214) cohorts were independently established. The primary outcome was time-to-recurrence after LT. A MoRAL-AI was derived from the derivation cohort with a residual block-based deep neural network. The median follow-up duration was 74.7 months (interquartile-range, 18.5–107.4); 204 patients (36.2%) had HCC beyond the Milan criteria. The optimal model consisted of seven layers including two residual blocks. In the validation cohort, the MoRAL-AI showed significantly better discrimination function (c-index = 0.75) than the Milan (c-index = 0.64), MoRAL (c-index = 0.69), University of California San Francisco (c-index = 0.62), up-to-seven (c-index = 0.50), and Kyoto (c-index = 0.50) criteria (all *p <* 0.001). The largest weighted parameter in the MoRAL-AI was tumor diameter, followed by alpha-fetoprotein, age, and protein induced by vitamin K absence-II. The MoRAL-AI had better predictability of tumor recurrence after LT than conventional models. The MoRAL-AI can also evolve with further data.

## 1. Introduction

Liver transplantation (LT) can be the most effective treatment among all treatment options for hepatocellular carcinoma (HCC) in carefully selected patients who meet certain criteria. Although the Milan criteria (MC) were introduced in 1996, they are still the most widely used system [1]. While the MC have been extensively used, recent data suggest that the MC may be too conservative when selecting an LT candidate [2]. Therefore, several other models have been developed, such as the University of California San Francisco (UCSF), up-to-seven, and the Kyoto criteria and model, to predict tumor recurrence after living donor LT (MoRAL) scores [3,4,5,6]. However, there are still no standard criteria. Moreover, due to continuous advances in diagnostic and therapeutic techniques for HCC, the establishment of standard criteria for LT becomes increasingly difficult.

Several factors related to HCC recurrence were identified in previous models, including maximum tumor diameter, tumor number, portal vein invasion, and serum tumor markers such as alpha-fetoprotein (AFP) and protein induced by vitamin K absence-II (PIVKA-II). When these factors were applied to previous models, a cutoff value of each factor was determined by conventional statistical methods. However, each factor was applied to the models as binary data, despite comprising continuous values. Models relying on binary data are simple and intuitive but may have inferior accuracy to ones built with continuous data. Moreover, the results of each previous model were also dichotomous: HCC recurrence or no recurrence after LT. It would be more helpful for precise medical decisions if the risk of HCC recurrence was presented as a continuous recurrence probability according to follow-up duration rather than as a simple dichotomous conclusion.

With recent advances in artificial intelligence (AI) including deep learning technology, this approach has also been applied to the medical field, particularly in diagnostic algorithms based on medical images such as those of radiology or histology [7,8]. There are also attempts to support clinical decision making using AI [9]. However, deep learning technology has not yet been applied to predict a patient’s future prognosis, because a prediction algorithm is much more complicated than a diagnostic algorithm and numerous parameters need to be considered. Therefore, this study aimed to develop and validate a novel model to predict tumor recurrence after LT by applying AI (MoRAL-AI) to patients with HCC.

## 2. Methods

### 2.1. Study Participants

This multicenter study included consecutive patients who underwent living donor LT at three large-volume centers in Korea. A derivation cohort (eligible patients from Samsung Medical Center [SMC] and National Cancer Center [NCC] from June 2003 to July 2013) and validation cohort (eligible patients from Seoul National University Hospital [SNUH] from September 2001 to January 2013) were established. All centers have extensive experience with LT (100–200 cases per year and >20 years of experience) and have reported similar post-LT survival and HCC recurrence rates.

The clinical data used to develop the model were based on pre-transplantation data. Before LT, HCC was mainly diagnosed using imaging-based diagnostic protocols according to international guidelines [10,11,12,13], although the diagnosis was histologically confirmed in all patients after LT. The patients without histologically confirmed HCC in post-LT tissues were excluded. The tumor burden was evaluated through pre-transplantation imaging results and laboratory findings. Factors for developing new models were selected based on previous models along with age and sex: maximum tumor diameter, tumor number, AFP, PIVKA-II, and portal vein invasion [1,3,4,5]. Patients who lacked clinical, radiological, or laboratory data within 1 month before LT were excluded, since the deep learning method could only be applied with complete data integrity. Patients with any evidence of extrahepatic metastasis of HCC were also excluded. To confirm extrahepatic spread, chest computed tomography (CT), bone scan, and 18F-FDG positron emission tomography/CT were performed.

In the case of portal vein tumor thrombosis, the peeling-off technique was used during LT [14], Immunosuppressive therapy after LT was implemented with the same agents: a combination of calcineurin inhibitor, mycophenolate mofetil, and corticosteroid. To prevent hepatitis B virus (HBV) reactivation in HBV-related HCC patients before LT, hepatitis B immunoglobulin and antiviral agents were administered according to international protocols [15,16]. Surveillance for HCC recurrence after LT was performed periodically through contrast-enhanced dynamic CT scans or magnetic resonance imaging every 2–4 months for the first 2 years after LT and every 3–6 months thereafter. HCC was diagnosed according to international guidelines [10,11,12,13].

This present study was conducted following the ethical guidelines of the World Medical Association Declaration of Helsinki. The protocol of this study was approved by the Institutional Review Board of each center, and the requirement for informed consent from patients was waived owing to the retrospective nature of the study (H-1706-109-859).

### 2.2. Statistical Analysis

Baseline characteristics of each data set were presented as mean ± standard deviation for normally distributed continuous variables and presented as the median with interquartile range for continuous variables with a skewed distribution. Discrete variables were noted by the number of subjects with percentages. To compare baseline characteristics between data sets, we used the Student’s *t*-test. The distribution of categorical variables was compared using the Fisher’s exact or chi-square test, as appropriate. Tumor recurrence time was calculated as the time from LT to detection of tumor recurrence after LT in a regular follow-up. Survival analysis was conducted using Kaplan-Meier survival analysis to estimate the cumulative tumor recurrence rate according to the risk group. To compare the prognostication power for tumor recurrence after LT among the models, we analyzed both discrimination and calibration performance using the concordance (c)-index for discrimination function and the Hosmer-Lemeshow test for calibration function. Statistical analyses were conducted using SPSS 22.0 (SPSS Inc., Chicago, IL, U.S.A.), SAS ver. 9.4 (SAS Institute, Cary, NC, USA), and R language version 3.6.3 (R Foundation for Statistical Computing, Vienna, Austria).

### 2.3. Deep Learning for Development of a Novel Prediction Model

The novel prediction model for HCC recurrence in post-LT patients was developed with a deep neural network (DNN). Among the selected factors, sex and portal vein invasion were Boolean (binary) data types; maximum tumor diameter, tumor number, AFP, PIVKA-II, and age were numerical (continuous) data types. To maintain and maximize the influence of each numerical factor, the model was developed by using the continuous numerical values themselves rather than by categorizing them with certain cutoffs. Since the related factors comprised two data types, we used DNN and arranged it so that each input variable was allocated its own input node, eliminating the influence of data type. In addition, numerical variables were normalized before their inclusion in the neural network.

To improve the model’s predictive performance, we used the residual learning framework from ResNet architecture, which presented good performance in a previous image recognition approach [17]. We applied residual learning to each stacked layer in this model. Residual learning is established through the connections between stacked layers. Integration of the subsequent input and output variables of each layer can provide additional nonlinearity and reduce the additionally generated weight to improve learning performance. We applied shortcuts (or skip connections) to increase the data learning performance by minimizing the data loss from centering layer responses, gradients, and propagated errors, implemented by shortcut connections.

The DNN was developed using TensorFlow (version 1.13; Google, Mountain View, CA, USA) and we used the Adam optimizer algorithm to optimize the proper model. We implemented the weighted cross entropy as a loss-function to control the class imbalance in the derivation set. The parametric rectified linear unit was used as an activation function and we adopted batch normalization with a minimal dropout. The learning rate was set at 1 × 10^−5^.

## 3. Results

### 3.1. Baseline Characteristics of the Derivation and Validation Cohorts

A total of 563 patients underwent living donor LT for HCC treatment at three study centers during the aforementioned study period; 349 patients were enrolled in the derivation cohort and 214 in the validation cohort. The baseline characteristics were evaluated through pre-transplantation imaging results and laboratory findings. The mean ages were 55.7 years in the derivation cohort and 53.7 years in the validation cohort. The proportions of men were 81.1% (286 of 349) in the derivation cohort and 85.0% (182 of 214) in the validation cohort. The proportions of beyond-MC patients were 32.7% in the derivation cohort and 42.1% in the validation cohort. Multi-nodularity of HCC was the most frequent reason for a beyond-MC status (46.5% in the derivation cohort and 51.1% in the validation cohort), followed by portal vein invasion and large HCC (>5 cm). The median follow-up duration was 71.4 months in the derivation cohort and 77.3 months in the validation cohort (Table 1).

### 3.2. Deep Learning Based Model

The optimal model consisted of only two residual blocks with seven layers because the size of the available data set was relatively small (Appendix A). Through performance assessment in the validation cohort, the MoRAL-AI (c-index, 0.75; 95% confidence interval (CI), 0.67–0.83) showed a significantly better discrimination function for predicting HCC recurrence compared with the Milan (c-index, 0.64; 95% CI, 0.60–0.67), MoRAL (c-index, 0.69; 95% CI, 0.59–0.79), UCSF (c-index, 0.62; 95% CI, 0.52–0.72), up-to-seven (c-index, 0.50; 95% CI, 0.40–0.59), and Kyoto (c-index, 0.50; 95% CI, 0.40–0.59) criteria (all *p <* 0.001) (Table 2). The calibration function was confirmed by the Hosmer-Lemeshow test (*p >* 0.05). Examples of the application of the model are provided in the HCC recurrence nomogram (Figure 1). According to clinical, radiological, and laboratory pre-transplantation data, the risk of HCC recurrence was stratified.

Although this MoRAL-AI model can provide a continuous probability of tumor recurrence but not specific cutoff values for risk stratification, three groups were established to confirm the model performance regarding the prediction of tumor recurrence probabilities. The high-risk group (5-year recurrence probability ≥ 0.5: hazard ratio (HR), 7.59; 95% CI, 3.63–15.88; *p <* 0.001) and moderate-risk group (0.2 ≤ 5-year recurrence probability < 0.5: HR, 2.41; 95% CI, 1.12–5.19; *p =* 0.025) showed significantly higher risk of tumor recurrence than the low-risk group (5-year recurrence probability < 0.2) in the validation cohort. The high-risk group also showed a significantly higher tumor recurrence rate than the moderate-risk group in the validation cohort (HR, 3.10; 95% CI, 1.61–6.01; *p =* 0.001). The expected HCC recurrence rate matched the observed tumor recurrence rates (Figure 2). The expected probabilities of tumor recurrence at 1, 3, and 5 years were 5.7%, 9.8%, and 9.8% in the low-risk group; 19.5%, 22.1%, and 23.5% in the moderate-risk group; and 46.8%, 53.2%, and 53.2% in the high-risk group, respectively. The MoRAL-AI model is shown as a web application (http://deeppacs.deepnoid.com:9000/APP/DEEP-LIVER/index.html#/). If a clinician inputs the pre-LT data of a specific patient in this website, the expected probabilities of HCC recurrence at the indicated time points after LT can be obtained. In addition, pre-LT data and post-LT recurrence data from centers around the world can be accumulated through this website after an adequacy review, which would further improve the performance of the current MoRAL-AI model.

### 3.3. Evaluation of the Weight of Each Factor in the DNN Model

To verify the importance of the respective factors in the MoRAL-AI model, we calculated the decline in the performance of each model by removing factors one-by-one. The performance of the MoRAL-AI model using all factors was assumed to be 100% and the importance of a specific factor was calculated by the c-index reduction in the DNN model developed without the specific factor. The c-indexes were calculated without each factor in turn as follows: without maximum tumor diameter (c-index = 0.62, 95% CI = 0.51–0.73), without AFP (c-index = 0.63, 95% CI = 0.53–0.73), without age (c-index = 0.64, 95% CI = 0.54–0.74), without PIVKA-II (c-index = 0.65, 95% CI = 0.55–0.77), without portal vein invasion (c-index = 0.66, 95% CI = 0.56–0.76), and without tumor number (c-index = 0.67, 95% CI = 0.57–0.77) (Appendix A). The largest weighted factor in the MoRAL-AI model was maximum tumor diameter, followed by AFP, age, and PIVKA-II. Tumor number was the least important factor. The relative importance of each factor is presented in Figure 3.

### 3.4. Performance Differences among the DNN Models according to the Inclusion of Factors 

To verify the performance of the optimal DNN model, we created several additional DNN models by including factors according to two aspects of tumor burden: imaging-based tumor burden (maximum tumor diameter, tumor number, and portal vein invasion) and biochemical tumor burden (AFP and PIVKA-II). A comparison of the performances of the DNN models revealed that the optimal DNN model (i.e., the model with the best performance) was made with all of the related factors (c-index, 0.75; 95% CI, 0.66–0.84; sensitivity, 0.76; specificity, 0.46). The performance of each DNN model worsened when the models were configured without either the imaging-based tumor burden (maximum tumor diameter, tumor number, and portal vein invasion) or biomaterial number (AFP and PIVKA-II) (the models without maximum tumor diameter and tumor number: c-index = 0.62, 95% CI = 0.51–0.72, sensitivity = 0.63, specificity = 0.62; the models without AFP and PIVKA-II: c-index =0.63, 95% CI = 0.53–0.72, sensitivity = 0.68, specificity = 0.59). When the DNN model was developed using only two factors of the maximum tumor diameter and tumor number, same as the MC, the c-index of the DNN model was 0.64 (95% CI, 0.55–0.73; sensitivity, 0.65; specificity, 0.47), which was similar to that of the conventional MC (Figure 4).

## 4. Discussion

In this study, we developed and validated a novel prediction model, called MoRAL-AI, for tumor recurrence after LT in patients with HCC. To the best of our knowledge, this is the first prediction model based on deep learning algorithms. The performance of MoRAL-AI was confirmed by using an independent validation cohort and was better than that of the MC, currently the most widely used criteria, as well as other prediction models. The MoRAL-AI is served through the website, and it can evolve with further data accumulation from various cohort groups. With this evolution, we can establish more evolved criteria of LT.

For LT as a treatment option for HCC, the underlying hepatic function of the recipient is not an important factor. While the recipient usually has liver cirrhosis before LT, the recipient liver is completely replaced by the donor liver and the severity of the pre-LT cirrhosis might thus not affect the post-LT tumor recurrence [18]. Therefore, predictive factors that are related to the post-LT HCC recurrence might include only the tumor burden and the biological aggressiveness of tumor cells before LT. Imaging studies can provide tumor-related information (e.g., tumor number, maximum tumor diameter, and vascular tumor invasion). On the other hand, serum levels of tumor markers (AFP and PIVKA-II) can reflect both tumor burden and biological aggressiveness because AFP and PIVKA-II had a significantly positive correlation with histological aggressive findings (microvascular invasion, perineural invasion, and serosal invasion) as well as tumor burden. Among previous models, the MC, UCSF, and up-to-seven models comprise only the factors related to imaging-based tumor burden, whereas the MoRAL score consists of only serum tumor markers [1,3,5,6]. The Hangzhou criteria and Kyoto criteria consider both imaging-based tumor burden and tumor markers [4,19]. The MoRAL-AI was developed based on both imaging-based tumor burden and biochemical tumor markers to maximize its performance.

The general disadvantage of the deep learning method is that it typically requires a large amount of data; it has previously been applied to the analysis of medical images such as plain X-ray, CT, or histology images, whose data can be abundantly obtained since a number of images are being taken during daily clinical practice [20,21]. However, when a specific disease is being analyzed, the size of the available data set is generally limited. For example, in Korea, only 1400 cases of LT were performed in 2015 [22], whereas approximately 265,000,000 plain X-rays were performed [23]. Moreover, it is generally complicated to collect demographic information and clinical results via a unified form due to the different data forms used by hospitals. Thus, data from fewer than 1000 cases have been analyzed with conventional statistical methods that can evaluate a relatively small dataset. However, our current model was derived from a relatively small derivation cohort containing 349 patients and showed a better predictive power than previous models. This result might suggest that DNN models can be developed even with relatively few data points if the potential prognostic factors for the prediction model have previously been well identified. Indeed, the predictive factors used in this study, such as tumor diameter [1,3,4,5,19], tumor number [1,3,4,5], AFP [6,19], PIVKA-II [4,6], and portal vein invasion [1,3,4,5,19,24], were well-identified parameters in previous studies.

The DNN model has several strengths compared with conventional statistical modeling. First, by applying a deep learning method to the prediction model, it is possible to derive more accurate continuous probability results. In the conventional statistical method, a scoring system is established to divide risk groups and continuous variables are stratified according to arbitrary cutoffs. However, DNN models can use continuous data rather than converting them into categorical or binary variables. Thus, they can provide more accurate and individualized results, which means that they can calculate the individual tumor recurrence probability at any time point after an LT according to baseline clinical information. Second, because DNN models were originally designed to be calculated by computer, there is no need to focus on ease of use. In contrast, previous conventional models had to be intuitive and easy to use and simple models comprising fewer factors were thus preferred. However, because DNN models involve an automatic calculation on a web application, there is no need to limit the number of factors or to consider the complexity of the formulae. Third, DNN models can evolve continuously with further data accumulation [25]. Previous models, such as the MC system which was developed in 1996, have not been changed, despite the accumulation of new data. However, DNN models can continuously improve their performances through additional data training.

This study has several limitations. First, it is impossible to understand the outcome operations resulting from deep learning. This is a general shortcoming of deep learning methods. Second, because this model was developed from Asian patients who underwent living donor LTs and whose underlying liver diseases is predominantly chronic hepatitis B, further validation in Western countries and deceased donor LT cohorts is warranted. Our model can provide additional options to select for deceased or living donor LT in our web application with further validation. Third, PIVKA-II is generally measured in Asian countries before LT, but is less commonly measured in Western countries. With further data training with other data sets, it may be possible to develop another model with high performance power without a certain factor like PIVKA-II.

## 5. Conclusions

In conclusion, the MoRAL-AI presented better performance in the prediction of HCC recurrence after LT than previous models. It can be served in real time through the website. Moreover, the MoRAL-AI can evolve with further data accumulation from various cohort groups. With this evolution, we can establish a standard criteria of LT.

## Figures and Tables

**Figure 1 cancers-12-02791-f001:**
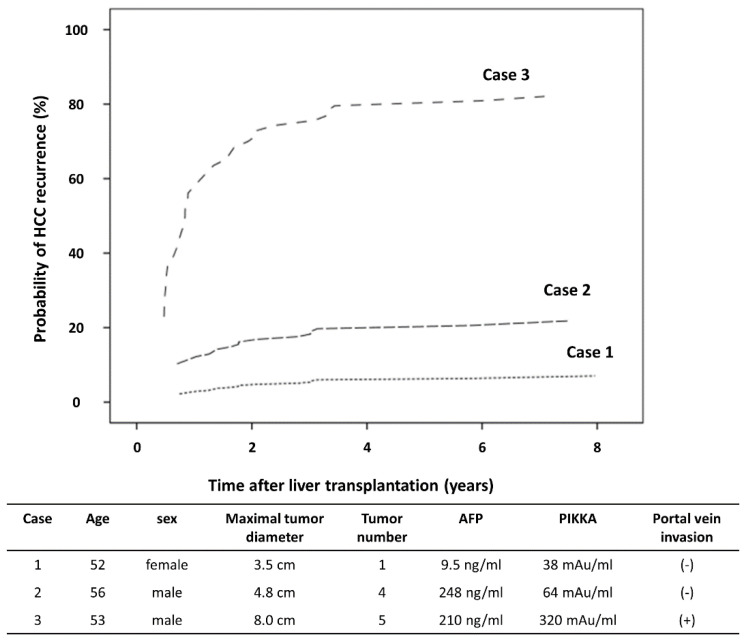
The HCC recurrence nomogram. The expected HCC recurrence rates of 3 hypothetical patients were presented, according to clinical, radiological, and laboratory data of pre-transplantation.

**Figure 2 cancers-12-02791-f002:**
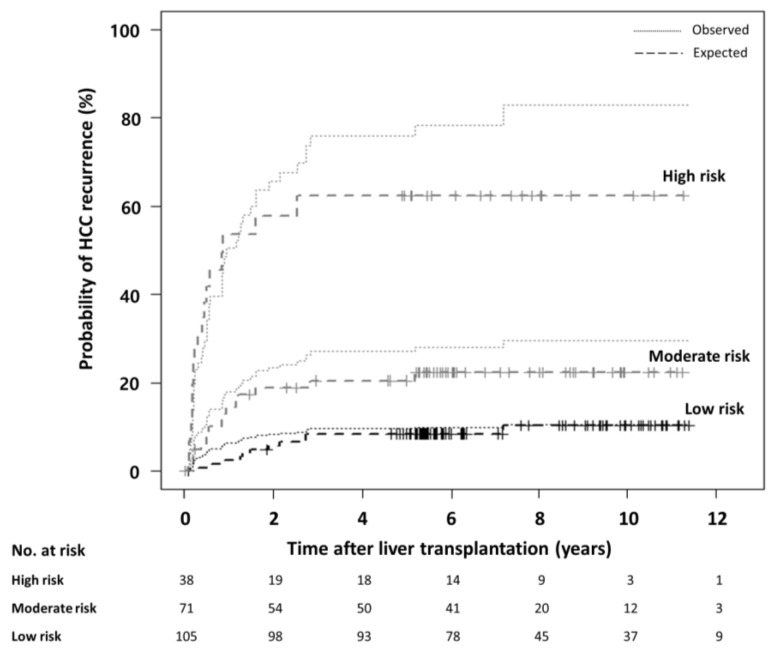
The expected versus the observed HCC recurrence after liver transplantation according to the MoRAL-AI in the validation cohort. To confirm the model performance, three groups which were divided by three tumor recurrence probabilities were compared. Among three groups, the risk of tumor recurrence differed significantly. The dashed lines represent the observed HCC recurrence and dotted lines represent the expected HCC recurrence.

**Figure 3 cancers-12-02791-f003:**
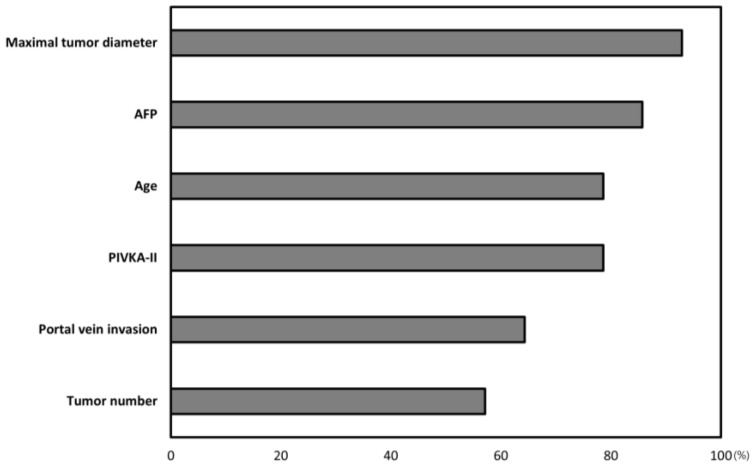
Weight of each prognostic factor in MoRAL-AI. The largest weighted factor in MoRAL-AI was tumor diameter, followed by AFP (alpha-fetoprotein), age, and PIVKA-II (protein induced by vitamin K absence-II).

**Figure 4 cancers-12-02791-f004:**
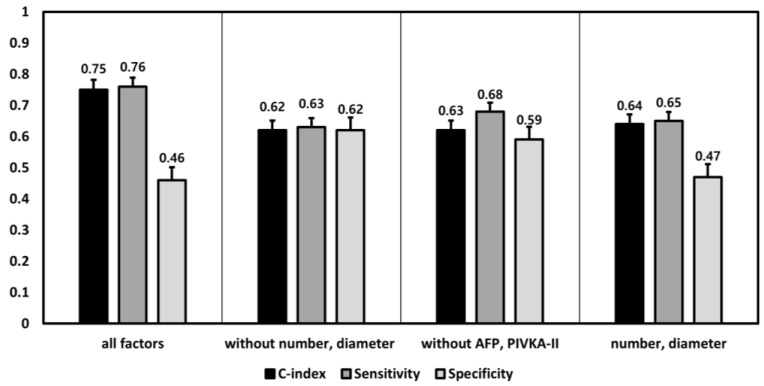
The performance differences among the DNN (deep neural network) models according to the changes of including factors in deep learning. An additional several DNN models were developed and evaluated, according to including factors which were divided by two aspects of the tumor burden: image-based tumor burden vs. biochemical tumor burden.

**Table 1 cancers-12-02791-t001:** Baseline patient characteristics based on pre-transplantation examination.

Patients Characteristics	Derivation Set (*n* = 349)	Validation Set (*n* = 214)	*p*-Value
Age, years	55.7 ± 7.8	53.7 ± 7.6	0.004 ^†^
Sex, Male, (*n*, %)	286 (81.1%)	182 (85.0%)	0.229 ^‡^
Maximal Tumor size, cm	3.0 ± 2.1	2.6 ± 2.1	0.021 ^†^
Tumor number, no.	1.0 (1.0–3.0)	2.0 (1.0–3.0)	0.046
AFP, ng/mL	19.0 (7.1–130.5)	13.9 (5.5–70.6)	0.060
PIVKA-II, mAU/mL	29.0 (15.0–137.5)	31.5 (19.0–111.8)	0.171
Portal vein invasion, (*n*, %)	44 (12.6%)	39 (18.2%)	0.068 ^‡^
BCLC stage	40/156/65/33/55	21/79/34/26/54	0.042 ^‡^
0/A/B/C/D, (*n*, %)	11.5/44.7/18.6/9.5/15.8	9.8/36.9/15.9/12.1/25.2	
Type of HCC	279/25/0	192/20/2	0.214 ^‡^
Nodular/diffuse or infiltrative, (*n*, %)	91.8/8.2/0.0	89.2/9.3/0.9	
Median follow-up, months	71.4 (12.8–104.3)	77.3 (56.0–117.7)	0.004
Proportion of beyond-MC	114 (32.7%)	90 (42.1%)	0.024 ^‡^
Portal vein invasion, (*n*, %)	44 (38.6%)	39 (43.3%)	0.494 ^‡^
Multinodular HCCs (≥4), (*n*, %)	53 (46.5%)	46 (51.1%)	0.512 ^‡^
Large HCCs (>5 cm), (*n*, %)	38 (33.3%)	20 (22.2%)	0.081 ^‡^

AFP, alpha-fetoprotein; PIVKA-II, protein induced by vitamin K absence-II; BCLC, Barcelona Clinic Liver Cancer; HCC, hepatocellular carcinoma; MC, Milan criteria. Note. Data are expressed as *n* (%), mean ± standard deviation for normally distributed continuous variables, or the median with interquartile range for continuous variables with a skewed distribution. ^†^ by student *t*-test; ^‡^ by Pearson’s Chi-square; Mann-Whitney test.

**Table 2 cancers-12-02791-t002:** C-indices for predicting HCC (hepatocellular carcinoma) recurrence by various models in validation cohort.

Model	c-Index	95% Confidence Interval		*p*-Value ^†^
		Lower	Upper	
MoRAL-AI	0.75	0.67	0.83	Ref.
Milan criteria	0.64	0.60	0.68	<0.001 ^†^
MoRAL	0.69	0.59	0.79	<0.001 ^†^
UCSF	0.62	0.52	0.72	<0.001 ^†^
Up-to-seven	0.50	0.40	0.59	<0.001 ^†^
Kyoto criteria	0.50	0.40	0.59	<0.001 ^†^

UCSF, University of California, San Francisco. ^†^ Compare to the c-index of the MoRAL-AI.

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
