# Peer review of "Novel Model to Predict HCC Recurrence after Liver Transplantation Obtained Using Deep Learning: A Multicenter Study"

_cancers, 2020, doi:10.3390/cancers12102791_

Round 1

Reviewer 1 Report

Dear Editor, The manuscript submitted by Nam et al. is a new predictive model for HCC recurrence after liver transplantation. 

the authors developed a model using the deep learning with a multicentric database of 563 patients. 

The MoRAL-AI model can predict the recurrence and divided patients into three groups: High,Moderate and, Low risk. 

A website is proposed for users. We need to have some easy preoperative data such age, Maximal Tumor Diameter, Tumor Number, Portal Vein Invasion, AFP and PIVKA-II. 

This is certainly an interesting predictive model. I have some questions and suggestions for the authors.

The first and main limitation is that the model is created on a living donor liver transplantation cohort. This undoubtedly changes the outcomes of patients transplanted for HCC.  The over 25 years of debate of Milan Criteria and the need to find an easy model for all patients and countries are still ongoing. 

I suggest to perform a validation on a different cohort or to include DCdonor liver transplantation. Moreover, PIVAK-II is used in Eastern but less frequent for the Western countries. To work a model need to be used worldwide in an easy way. I'm afraid that in this way the manuscript will be not used by the liver transplant community. 

Author Response

Reviewer #1:

1) Reviewer #1’s Comment #1: The first and main limitation is that the model is created on a living donor liver transplantation cohort. This undoubtedly changes the outcomes of patients transplanted for HCC.  The over 25 years of debate of Milan Criteria and the need to find an easy model for all patients and countries are still ongoing. 

Author’s response: Thank you for your insightful comment. As you pointed out, our model was developed based on living donor liver transplantation cohort. Although the differences of HCC recurrence between deceased donor liver transplantation (DDLT) and living donor liver transplantation (LDLT) cohort have been reported in previous studies, there have been also several studies reporting no significant difference in outcomes including risk of HCC recurrence between two cohorts. As we agree with the reviewer's opinion, we hope to extend our study to patients who underwent DDLT for HCC in the future study. In addition, we will add an additional option selecting DDLT and LDLT in our web application. Discussion section has been revised as follows:

“Second, because this model was developed from Asian patients who underwent living donor LTs and whose underlying liver diseases is predominantly chronic hepatitis B, further validation in Western countries and deceased donor LT cohorts is warranted. Our model can provide additional selecting option for DDLT and LDLT in our web application with further validation.”

(lines 297–298)

2) Reviewer #1’s Comment #2: I suggest to perform a validation on a different cohort or to include DC donor liver transplantation. Moreover, PIVAK-II is used in Eastern but less frequent for the Western countries. To work a model need to be used worldwide in an easy way. I'm afraid that in this way the manuscript will be not used by the liver transplant community. 

Author’s response: Thank you for your kind comment. We agree to your concern regarding PIVKA-II. We can develop the derivative another variant model utilizing variables except for PIVKA-II by using same algorithm. However, at this time, the performance of the variant model without PIVKA-II was relatively low. If the cohorts from Western countries where the PIVKA-II cannot be easily measured are available, our model can be easily modified to utilize only available factors. We have added this limitation in the Discussion section, as follows:

Third, PIVKA-II is generally measured in Asian countries before LT, but is less commonly measured in the Western countries. With further data training with other data sets, it may be possible to develop another model with high performance power without a certain factor like PIVKA-II.” 

(lines 298–301)

Reviewer 2 Report

In this study, the authors aim to develop and validate a novel model to predict tumor recurrence after liver transplantation (LT) by applying AI (MoRAL77 AI) to patients with hepatocellular carcinoma (HCC). The authors demonstrate that the MoRAL-AI presented better performance in the prediction of HCC recurrence after LT than previous models. The studies are nicely executed, and most of the findings are straight forward. While the concept of the work is very interesting and informative, there are a few concerns that the authors need to address in their study. 

  1. Please provide significance of the research in more details.
  2. Please improve discussion section.
  3. Please cite the following articles in your manuscript and include in reference section
    1. Transglutaminase-2 facilitates extracellular vesicle-mediated establishment of the metastatic niche
    2. Spleen Tyrosine Kinase–Mediated Autophagy Is Required for Epithelial–Mesenchymal Plasticity and Metastasis in Breast Cancer.
    3. Autocrine fibronectin inhibits breast cancer metastasis.
    4. Pyruvate carboxylase supports the pulmonary tropism of metastatic breast cancer
    5. Regulation of epithelial-mesenchymal transition and metastasis by TGF-β, P-bodies, and autophagy
    6. The Dynamic Relationship of Breast Cancer Cells and Fibroblasts in Fibronectin Accumulation at Primary and Metastatic Tumor Sites
    7. Dynamic transition of the blood-brain barrier in the development of non-small cell lung cancer brain metastases
    8. Inhibition of pyruvate carboxylase by 1α, 25-dihydroxyvitamin D promotes oxidative stress in early breast cancer progression

Author Response

Reviewer #2:

1) Reviewer #2’s Comment#1: In this study, the authors aim to develop and validate a novel model to predict tumor recurrence after liver transplantation (LT) by applying AI (MoRAL- AI) to patients with hepatocellular carcinoma (HCC). The authors demonstrate that the MoRAL-AI presented better performance in the prediction of HCC recurrence after LT than previous models. The studies are nicely executed, and most of the findings are straight forward. While the concept of the work is very interesting and informative, there are a few concerns that the authors need to address in their study. 

  1. Please provide significance of the research in more details.
  2. Please improve discussion section.
  3. Please cite the following articles in your manuscript and include in reference section
    1. Transglutaminase-2 facilitates extracellular vesicle-mediated establishment of the metastatic niche
    2. Spleen Tyrosine Kinase–Mediated Autophagy Is Required for Epithelial–Mesenchymal Plasticity and Metastasis in Breast Cancer.
    3. Autocrine fibronectin inhibits breast cancer metastasis.
    4. Pyruvate carboxylase supports the pulmonary tropism of metastatic breast cancer
    5. Regulation of epithelial-mesenchymal transition and metastasis by TGF-β, P-bodies, and autophagy
    6. The Dynamic Relationship of Breast Cancer Cells and Fibroblasts in Fibronectin Accumulation at Primary and Metastatic Tumor Sites
    7. Dynamic transition of the blood-brain barrier in the development of non-small cell lung cancer brain metastases
    8. Inhibition of pyruvate carboxylase by 1α, 25-dihydroxyvitamin D promotes oxidative stress in early breast cancer progression

Author’s response: Thank you for your helpful comments. Following your recommendations, we revised the manuscript as follows:

“In this study, we developed and validated a novel prediction model—called MoRAL-AI—for tumor recurrence after LT in patients with HCC. To the best of our knowledge, this is the first prediction model based on deep learning algorithms. The performance of MoRAL-AI was confirmed by using an independent validation cohort and was better than that of the MC, currently the most widely used criteria, as well as other prediction models. The MoRAL-AI is served through the website, and it can evolve with further data accumulation from various cohort groups. With this evolution, we can establish more evolved criteria of LT.

(lines 244–246)

The MoRAL-AI was developed based on both imaging-based tumor burden and biochemical tumor markers to maximize its performance.

(lines 260–261)

“This study has several limitations. First, it is impossible to understand the outcome operations resulting from deep learning. This is a general shortcoming of deep learning methods. Second, because this model was developed from Asian patients who underwent living donor LTs and whose underlying liver diseases is predominantly chronic hepatitis B, further validation in Western countries and deceased donor LT cohorts is warranted. Our model can provide additional selecting option for DDLT and LDLT in our web application with further validation. Third, PIVKA-II is generally measured in Asian countries before LT, but is less commonly measured in the Western countries. With further data training with other data sets, it may be possible to develop another model with high performance power without a certain factor like PIVKA-II.

(lines 297–301)

“In conclusion, the MoRAL-AI presented better performance in the prediction of HCC recurrence after LT than previous models. It can be served in real time through the website. Moreover, the MoRAL-AI can evolve with further data accumulation from various cohort groups. With this evolution, we can establish a standard criteria of LT.

(lines 305–306)

Reviewer 3 Report

In this paper, Dr Nam and Coworkers presented an original model adopting artificial intelligence to predict recurrent HCC after OLTx. The new model appeared more performing than others, such as the Milan, UCSF, Up-to-Seven and Kyoto criteria and is based on tumour diameter, AFP, age and PIVKA-II.

The proposed approach was very selective in terms of inclusion criteria and centres of excellence involved, but above all, it was based on a prognostic algorithm through a continuous recurrence probability according to FU duration.

The paper is clear, well articulated in its presentation and with a valid methodology developed through the deep neural network. The number and description of the 2 derivation and validation cohorts appear appropriate. Finally, the presentation of 3 groups with high, moderate or low risk of tumour recurrence based on tumour biological and biochemical parameters optimized the accuracy of the model.

I have no adjustments to propose, but only the suggestion of comparing and validating the model also in other populations of HCC patients who are undergoing liver transplantation, particularly in the EU and US.

Author Response

Reviewer #3:

1) Reviewer #3’s Comment#1: I have no adjustments to propose, but only the suggestion of comparing and validating the model also in other populations of HCC patients who are undergoing liver transplantation, particularly in the EU and US.

Author’s response: Thank you for your suggestion. Following your recommendations, we revised the manuscript as follows:

“In this study, we developed and validated a novel prediction model—called MoRAL-AI—for tumor recurrence after LT in patients with HCC. To the best of our knowledge, this is the first prediction model based on deep learning algorithms. The performance of MoRAL-AI was confirmed by using an independent validation cohort and was better than that of the MC, currently the most widely used criteria, as well as other prediction models. The MoRAL-AI is served through the website, and it can evolve with further data accumulation from various cohort groups. With this evolution, we can establish more evolved criteria of LT.

(lines 244–246)

“This study has several limitations. First, it is impossible to understand the outcome operations resulting from deep learning. This is a general shortcoming of deep learning methods. Second, because this model was developed from Asian patients who underwent living donor LTs and whose underlying liver diseases is predominantly chronic hepatitis B, further validation in Western countries and deceased donor LT cohorts is warranted. Our model can provide additional selecting option for DDLT and LDLT in our web application with further validation. Third, PIVKA-II is generally measured in Asian countries before LT, but is less commonly measured in the Western countries. With further data training with other data sets, it may be possible to develop another model with high performance power without a certain factor like PIVKA-II.

(lines 297–301)

Reviewer recommended several articles for citation, however, all recommended references were articles related to breast cancer. Therefore, we could not update our references section. The duplicate sentences pointed out by the editor were also corrected in revised manuscript.

Round 2

Reviewer 1 Report

The manuscript has been improved and can be now be accepted. 

Reviewer 2 Report

The authors have addressed all the comments.